# Oncogene-Addicted Non-Small-Cell Lung Cancer: Treatment Opportunities and Future Perspectives

**DOI:** 10.3390/cancers12051196

**Published:** 2020-05-08

**Authors:** Miriam Grazia Ferrara, Vincenzo Di Noia, Ettore D’Argento, Emanuele Vita, Paola Damiano, Antonella Cannella, Marta Ribelli, Sara Pilotto, Michele Milella, Giampaolo Tortora, Emilio Bria

**Affiliations:** 1Medical Oncology, Comprehensive Cancer Center, Fondazione Policlinico Universitario Agostino Gemelli IRCCS, Università Cattolica del Sacro Cuore, 00168 Rome, Italy; miriamgraziaferrara@gmail.com (M.G.F.); ettore.dargento@policlinicogemelli.it (E.D.); dr.emanuele.vita@gmail.com (E.V.); paoladamiano88@gmail.com (P.D.); cannella.antonella@libero.it (A.C.); marta.ribelli@gmail.com (M.R.); giampaolo.tortora@unicatt.it (G.T.); 2Medical Oncology, Università Cattolica del Sacro Cuore, 00168 Rome, Italy; vincedinoia@gmail.com; 3Medical Oncology, Oncologia medica, Humanitas Gavazzeni, 24125 Bergamo, Italy; 4Medical Oncology, Azienda Ospedaliera Universitaria Integrata, University and Hospital Trust of Verona, 37126 Verona, Italy; sara.pilotto@univr.it (S.P.); michele.milella@univr.it (M.M.)

**Keywords:** oncogene, addiction, EGFR, ALK, ROS1, lung

## Abstract

Before the introduction of tyrosine kinase inhibitors (TKIs) for a particular subgroup of patients, despite platinum-based combination chemotherapy, the majority of patients affected by non-small-cell lung cancer (NSCLC) did not live longer than one year. With deeper understanding of tumor molecular biology, treatment of NSCLC has progressively entered the era of treatment customization according to tumor molecular characteristics, as well as histology. All this information allowed the development of personalized molecular targeted therapies. A series of studies have shown that, in some cases, cancer cells can grow and survive as result of the presence of a single driver genomic abnormality. This phenomenon, called oncogene-addiction, more often occurs in adenocarcinoma histology, in non-smokers (except BRAF mutations, also frequent in smoking patients), young, and female patients. Several different driver mutations have been identified and many studies have clearly shown that upfront TKI monotherapy may improve the overall outcome of these patients. The greater efficacy of these drugs is also associated with a better tolerability and safety than chemotherapy, with fewer side effects and an extremely good compliance to treatment. The most frequent oncogene-addicted disease is represented by those tumors carrying a mutation of the epidermal growth factor receptor (EGFR). The development of first, second and third generation TKIs against EGFR mutations have dramatically changed the prognosis of these patients. Currently, osimertinib (which demonstrated to improve efficacy with a better tolerability in comparison with first-generation TKIs) is considered the best treatment option for patients affected by NSCLC harboring a common EGFR mutation. EML4-ALK-driven disease (which gene re-arrangement occurs in 3–7% of NSCLC), has demonstrated to be significantly targeted by specific TKIs, which have improved outcome in comparison with chemotherapy. To date, alectinib is considered the best treatment option for these patients, with other newer agents upcoming. Other additional driver abnormalities, such as ROS1, BRAF, MET, RET and NTRK, have been identified as a target mirroring peculiar vulnerability to specific agents. Oncogene-addicted disease typically has a low early resistance rate, but late acquired resistance always develops and therefore therapy needs to be changed when progression occurs. In this narrative review, the state of art of scientific literature about targeted therapy options in oncogene-addicted disease is summarized and critically discussed. We also aim to analyze future perspectives to maximize benefits for this subgroup of patients.

## 1. Oncogene Addiction

Non-small-cell lung cancer (NSCLC) is the main cause of cancer-related deaths worldwide [1]. Before introducing tyrosine kinase inhibitors (TKIs), the overall survival (OS) of most patients with advanced NSCLC receiving platinum-based combination chemotherapy was less than one year [2]. The identification of molecular aberrations leading to tumor growth and survival, so-called oncogene-addicted tumors, has dramatically changed the treatment landscape of NSCLC.

In the 1970s, baseline performance status, extent of disease, and weight loss represented the main predictors of survival [3]. With deep understanding of genomic landscape in cancer, treatment of NSCLC has faced the era of therapy based on tumor molecular characteristics. This new knowledge allowed the development and clinical introduction of personalized molecular targeted therapies [4].

Lung cancers usually evolve through a multistage process that can extend over a period of decades. This process is driven by the progressive accumulation of mutations and epigenetic abnormalities leading to the expression of multiple genes with different functions [5]. Despite this complexity, cancer cell growth and survival can often be impaired by the inactivation of a single oncogene [6]. Each somatic alteration occurring in a single cancer cell may be classified according to its consequences for cancer development. Driver mutations confer a growth advantage to the cells and have been positively selected during the cancer evolution. This phenomenon, called oncogene addiction, provides a rationale for molecular targeted therapy [6,7,8]. Numerous driver mutations have been identified in NSCLC, predominantly epidermal growth factor receptor (EGFR) and KRAS mutations, and EML4-ALK translocations [9]. Alterations involving ROS1, BRAF, MET, and NRTK genes are less frequent but featured by a strong targeting potential.

Nevertheless, the vast majority (97%) of mutations occurring during oncogenesis are passengers. Passenger mutations have no direct effect in terms of selective growth advantage for cancer cells, but they may be associated with a clonal expansion when occurring concomitantly with a driver mutation [4]. These mutations are likely to have only minimal biological consequences on the disease process, but their potentiality and role remain poorly understood. Therefore, a cell that acquires a molecular *driver* has probably already passenger somatic mutations within its genome [10].

Oncogene-addicted disease has also been evaluated in terms of tumor mutational burden (TMB), an emerging candidate biomarker for immune checkpoint inhibitors efficacy in lung cancer. TMB is usually low in oncogene-addicted tumors and there is an inverse correlation between TMB and clinical benefit deriving from EGFR-TKIs as assessed by OS and time to treatment discontinuation (TTD) [11]. On the contrary, PD-L1 is generally high in EGFR-mutated NSCLC, but immunotherapy appeared to be less effective in this subgroup of patients, and treatment is often burdened by serious side effects (Table 1 and Table 2); [12,13,14,15,16,17,18,19,20,21,22,23]. In addition, in contrast to non-oncogene addicted disease, oncogene-addicted disease has a low early resistance rate, but late acquired resistance always develops (Table 3).

In both cases, cancer represents a dynamic disease and over time it becomes more and more heterogeneous. As a result of this heterogeneity, a tumor may include different clones of tumor cells harboring distinct molecular signatures and with differential levels of treatment sensitivity [24]. Cancer molecular heterogeneity, in particular co-occurring genetic alterations, could also explain the variable response to TKIs observed in oncogene-addicted disease. Concomitant gene alterations collaboratively act as co-drivers favoring tumor progression and drug resistance. As an example, TP53 mutation is the most frequent genetic alteration co-occurring in EGFR-mutant NSCLC. This gene encodes a tumor suppressor protein that is involved in DNA repair, metabolism, cell cycle arrest, apoptosis, and senescence [25]. Non-disruptive TP53 mutations (those preserving some functional properties of the protein) represent an independent prognostic factor of shorter OS in advanced NSCLC (13.3 versus 24.6 months; HR = 2.08; *p* < 0.001) [26]. Overall, the majority of recent studies have shown that TP53 mutations are associated with poorer OS in NSCLC patients and support the hypothesis that TP53 (and perhaps other tumor suppressor genes) may affect the efficacy of traditional targeted therapy in molecularly-addicted NSCLC patients by triggering cell proliferation and passing the oncogenic power of the EGFR pathway [27,28,29]. To summarize, both TP53 mutations and TMB may be considered predictors of TKIs efficacy in oncogene-addicted disease [11,27,29].

## 2. EGFR Mutations

Mutations in EGFR (either small in-frame deletions in exon 19, del19, or amino acid substitution (leucine to arginine at codon 858, L858R) clustered around the ATP-binding pocket of the tyrosine kinase domain) are present in 10–26% of NSCLC and are more frequent in the Asiatic population [30]. Studies on lung cancer cell lines and transgenic mice harboring EGFR mutations have shown the oncogenic potential of these mutations, with enhanced response to EGFR inhibitors [31].

Since 2005, some clinical trials evaluated the efficacy of EGFR inhibitors in patients experiencing progressive disease after chemotherapy, regardless of their mutational profile, suggesting a modest advantage versus placebo [32,33]. In the same period, first data demonstrated that a subgroup of patients with NSCLC has specific mutations in the EGFR gene, which correlate with clinical responsiveness to the tyrosine kinase inhibitor gefitinib [34]. Mok et al. compared head-to-head gefitinib versus carboplatin-paclitaxel in patients with untreated lung adenocarcinoma, non-smokers or ex-smokers, demonstrating an increase in progression-free survival (PFS) in patients treated with gefitinib, with the greater benefit in the subgroup of patients who were positive for EGFR mutation [35].

The mechanism of action of first-generation EGFR-TKIs (as gefitinib and erlotinib) is to reversibly block the activation of downstream signaling induced by EGFR through binding to the ATP-binding sites.

In 2009 gefitinib was the first EGFR TKI approved as first-line treatment of advanced NSCLC patients harboring activating EGFR mutations, as demonstrated in four major clinical trials. The Iressa Pan-Asia Study (IPASS) trial [36] was the first randomized clinical trial to compare EGFR-TKI with chemotherapy in patients affected by lung adenocarcinoma who were former light smokers or nonsmokers. A significant benefit in terms of PFS was reported in favor of gefitinib, while no difference in OS was found between both treatments groups [18.8 versus 17.4 months, HR 0.90; 95% confidence intervals (CI) 0.79–1.02; *p* < 0.109], although this result was likely confounded by the high proportion of patients in the control arm who received TKI as subsequent treatment. Interestingly, the PFS benefit was confirmed only in the EGFR mutation-positive subgroup (median PFS 9.5 versus 6.3 months, HR 0.48 [95% CI 0.36–0.64]; *p* < 0.001), while in the EGFR mutation-negative subgroup TKI was inferior to chemotherapy (median PFS 1.5 versus 5.5 months, HR 2.85 [95% CI 2.05–3.98]; *p* < 0.001). Thus, activating EGFR mutations were identified as a relevant predictive factor for response to gefitinib. The outcomes of the other three trials (First-SIGNAL, NEJ002, and WJTOG3405) [37,38,39] confirmed the IPASS results in EGFR-mutant NSCLC patients when comparing gefitinib with doublet chemotherapy as first-line therapy. In all these studies, rash and liver dysfunction were the most common adverse events (AEs) reported with grade 3–4 toxicities described in less than 30% of patients [37].

Erlotinib received approval as a first-line treatment for patients with EGFR mutation in 2011 EURTAC [40] and OPTIMAL trials [41,42] proved the superiority in terms of PFS of erlotinib over first-line platinum-based chemotherapy in NSCLC patients harboring EGFR mutation (13.1 versus 4.6 months; HR 0.16 [95% CI 0.10–0.26]; *p* < 0.0001). Skin rash and liver dysfunction were described as the most common AEs with a low rate of grade 3–4 toxicities (17%) [41].

Novel EGFR TKIs were developed in order to improve the outcome of *EGFR*-mutated NSCLC. In 2013 the Food and Drug Administration (FDA) and the European Medicines Agency (EMA) approved afatinib in first-line treatment of EGFR mutant patients. Afatinib is a second-generation, orally bioavailable, irreversible EGFR inhibitor that can modify conserved cysteine residues within the catalytic domains of EGFR, HER2, and ErbB-4. By an irreversible covalent bond, it blocks enzymatic activity of active ErbB receptor family members until the synthesis of new receptors, thus the inhibitory action of afatinib is longer than reversible EGFR TKIs [43]. LUX-lung 3 and LUX-lung 6 trials [44,45] were randomized studies that compared afatinib with cisplatin plus pemetrexed or cisplatin plus gemcitabine as first-line treatment in patients affected by lung adenocarcinoma harboring *EGFR* mutations. Afatinib significantly improved PFS and overall response rate (ORR) compared with chemotherapy. A retrospective pooled analysis of LUX-lung 3 and LUX-lung 6 demonstrated an OS benefit with afatinib only in the del19 group and not in the L858R mutation group, suggesting an intrinsic prognostic meaning of these two mutations [46]. In LUX-lung 3 and LUX-lung 6 trials, the overall frequency of patients with uncommon EGFR mutations was 11% and afatinib showed activity for the majority of them. Based on these results, in 2018 the FDA approved afatinib for uncommon EGFR mutant patients. The most common treatment-related AEs reported were diarrhea, rash/acne, stomatitis, and nail effects; AEs grade 3–4 occurred in 49% of patients [44].

Dacomitinib is a pan-HER TKI, able to target EGFR, ErbB2, and ErbB4 kinase, and it regulates the EGFR signaling pathway by irreversibly bind to the ATP-pocket [47,48]. In the ARCHER 1050 trial, dacomitinib significantly improves PFS and OS compared with gefitinib (34.1 versus 26.8 months, HR 0.76; *p* = 0.044). Although the ARCHER 1050 trial was conducted in the Asiatic population without brain metastases, dacomitinib is the only TKI that has shown superior OS in comparison with another TKI. This benefit was achieved at the cost of higher toxicity, especially dermatitis acneiform, diarrhea, and liver dysfunction (frequency of grade 3–4 adverse events 63% versus 41%) and a detrimental effect on quality of life (QoL) [49]. Based on ARCHER 1050 results, in 2019 the EMA approved dacomitinib in patients with EGFR del19 or 21L858R.

Multiple phase III trials comparing first- and second-generation EGFR TKIs with platinum-based chemotherapy as frontline treatment for patients affected by advanced NSCLC have been reported. A consistent benefit in favor of EGFR TKIs was observed across all studies in terms of PFS, ORR, and disease control rate. The median PFS with these compounds ranged from 8.0–13.1 months, compared with 4.6–6.9 months with chemotherapy [50].

With the exception of ARCHER 1050 trial with dacomitinib, in comparative studies, no relevant differences in terms of efficacy were observed between the other first- and second-generation EGFR TKIs. In CTONG 0901 [51] trial, patients were randomized to receive erlotinib or gefitinib. Median OS was similar (22.9 versus 20.1 months [95% CI 0.63–1.13], *p* = 0.250). In LUX-Lung 7 [52] trial, patients were randomized to receive afatinib or gefitinib. A median OS of 27.9 versus 24.5 months (HR = 0.86, 95% CI 0.66‒1.12; *p* = 0.2580) was reported, while PFS was significantly improved with afatinib (HR = 0.73, 95% CI 0.57‒0.95; *p* = 0.0165) although with a greater toxicity (grade  ≥ 3 adverse events 31.3% versus 19.5%).

To sum up, in the pre-osimertinib era, in the treatment of lung cancer patients with common *EGFR* mutations (L858R and exon del19), the choice of a first or second-generation EGFR TKI depends on toxicity profile, physician’s preference, and the local availability of each agent. With regard to metabolism, afatinib is not an inhibitor or an inducer of CYP450 enzymes. Therefore, in patients who need concomitant medications, afatinib therapy reduces the risk of drug-drug interactions. 

Despite high initial response and disease control rate, virtually all patients receiving these EGFR TKIs eventually experience tumor progression due to the emergence of drug resistance [53]. Resistance to TKIs is most commonly acquired de novo during treatment, but can also occur due to the outgrowth of pre-existing resistant subclones [24]. In approximately 50% of patients treated with first- and second-generation EGFR TKIs, resistance is mediated by the acquisition of the gatekeeper T790M mutation, which results in the sterical blockade of these TKIs binding and concurrently increases the kinase affinity for ATP [54]. T790M often coexisted with pre-existing sensitizing EGFR mutations, such as del19 or 21L858R, in particular with 21L858R rather than del19 [55].

In order to increase the awaited benefit in *EGFR*-mutant NSCLC patients, new TKIs were developed. Among them, osimertinib is an irreversible third-generation EGFR TKI that is active against T790M resistance mutation other than del19 and L858R sensitizing mutations. This TKI forms a covalent bond to the cysteine residue at position 797 and has lower activity than the aforementioned TKIs against wild-type EGFR protein, therefore improving its safety profile. 

In the AURA 3 randomized phase III trial [56], osimertinib significantly prolonged median PFS and ORRs than cisplatin plus pemetrexed in *T790M* positive NSCLC patients who had disease progression after first-line EGFR-TKI. In comparison with chemotherapy, patients receiving osimertinib had also a lower incidence of grade ≥3 adverse events (23% vs. 47%) and an improved QoL with better scores for lung cancer symptoms [56]. The extended benefit of the sequential administration of a first-generation EGFR TKI followed by osimertinib observed in this study led to the approval of this compound for patients with NSCLC harboring *T790M* mutation and experiencing disease progression after first-generation or second-generation EGFR TKIs. In this setting, blood-based and/or tumor sampling analysis is mandatory to determine the *T790M* mutational status after disease progression, owing to the clinical benefits that may be obtained from a sequential strategy with osimertinib [57,58,59]. Plasma circulating tumor-DNA (ctDNA) still cannot fully replace tissue to diagnose acquired T790M mutation after disease progression due to a lower sensitivity of the blood analysis [60]. Given the reduced risk of plasma analysis compared with an invasive biopsy procedure, data support a new paradigm for resistance management, with rapid plasma genotyping as a diagnostic option before undergoing a tumor biopsy. Patients negative for T790M in plasma, however, should undergo a biopsy to determine T790M status because of the risks of false-negative plasma results [61].

Wang et al., collected clinical characteristics of patients diagnosed with EGFR mutation and identified patients with primary and acquired T790M mutation. They found that primary T790M mutation appears to be basically refractory to older TKI. Conversely, osimertinib works with the same impact in patients with primary or acquired T790M [55]. Based on this evidence [62,63], it was suggested that osimertinib may also be an effective first-line therapy for patients with advanced EGFR mutated NSCLC. In the phase 3 FLAURA trial, osimertinib significantly improved PFS compared with standard EGFR-TKIs gefitinib and erlotinib as first-line treatment of *EGFR* mutated advanced NSCLC, with a similar safety profile and lower rate of serious adverse events [64]. Due to greater penetration of the BBB, osimertinib also reduced CNS progression events [64]. Furthermore, TTD and time to second progression were superior in osimertinib-treated patients and more patients received further lines of therapy [65]. The final OS analysis has shown that osimertinib improves OS in comparison with standard of care (median OS 38.6 [95% CI 34.5–41.8] versus 31.8 months [95% CI 26.6–36]; HR 0.799, *p* = 0.046) [66]. Comparing the recent randomized clinical trials with the FLAURA control arm, first- and second-generation TKIs show an inferior median OS in comparison with the osimertinib arm. Moreover, 30% of patients do not receive subsequent anti-cancer treatment both after first-line osimertinib and standard of care TKIs [64]. This evidence suggests that the best treatment should be used in the first-line setting, avoiding that proportion of patients that will not be treated with a second-line therapy. On the basis of all these data, osimertinib is highly recommended in the first-line setting of EGFR mutant advanced NSCLC thanks to its superiority in terms of PFS, OS, and intracranial efficacy with a manageable toxicity profile. Few data are available for uncommon EGFR mutations, but results are consistent with previous reports confirming osimertinib efficacy [67].

In AURA 3 and FLAURA trials mechanisms of resistance to osimertinib have been identified through plasma samples collected at baseline and following disease progression. The most common acquired resistance mechanism detected were MET amplification and EGFR C797S mutation (about 15% each); other mechanisms included HER2 amplification, PIK3CA, KRAS, NRTK, RET, and FGFR3 mutations (2–7%) [68] (Table 4).

To better control disease progression and improve clinical outcomes, many studies have evaluated the combination of TKI with other drugs. In a TATTON study, a phase Ib trial, MET-amplified, and EGFR-mutant patients were treated with osimertinib plus savolitinib showing an acceptable risk-benefit profile and encouraging anti-tumor activity of the combination (objective partial responses in 48% of patients) [69]. In a RELAY trial, patients were randomly assigned to receive erlotinib plus ramucirumab or erlotinib plus placebo showing that PSF was significantly longer in the ramucirumab plus erlotinib group (19.4 months (95% CI 15.4–21.6)) than in the placebo plus erlotinib group (12.4 months (95% CI 11.0–13.5)), with HR = 0.59 (95% CI 0.46–0.76; *p* < 0.0001). Grade 3–4 treatment-emergent adverse events were reported in 72% in the ramucirumab plus erlotinib group versus 54% in the placebo plus erlotinib group [70]. Many trials have evaluated the combination of erlotinib plus bevacizumab versus erlotinib plus placebo but, to date, there is no evidence of OS benefit (Table 4). Norohna et al. have demonstrated that a combination of gefitinib plus pemetrexed/carboplatin significantly prolonged PFS and OS but increased toxicity in EGFR-mutant patients with advanced NSCLC [71].

Nowadays, it is clear that many efforts and steps forward have been made to improve clinical outcomes without worsening quality in this subgroup of patients, but there is still a long way to go.

## 3. ALK Translocations

Rearrangements of the anaplastic lymphoma kinase (ALK) gene are present in 3% to 5% of NSCLC [72] and typically occur in younger patients who have never smoked or have a history of light smoking and that have adenocarcinoma histotype [73].

The identification of ALK translocation in NSCLC was reported for the first time in 2007, when in some preclinical models it was demonstrated that the chemical inhibition of ALK fusion protein led to a dramatic reduction in tumor cell growth and to a significant survival advantage of ALK-positive lung cancer murine xenografts [72]. The most frequent ALK translocations usually involve the gene coding for the Echinoderm Microtubule-associated protein Like 4 (*EML4*) [74,75]. After dimerization, EML4-ALK transcript constitutively triggers Mitogen-Activated Protein Kinase (MAPK), Janus Kinase with Signal Transducer And Activator Of Transcription (JAK-STAT) and Phosphoinositide-3-Kinase with VAkt Murine Thymoma Viral Oncogene Homolog (PI3K-AKT), leading to an increase in proliferation and survival and boosting the angiogenetic switch in cancer cells [76,77].

Besides EGFR, the discovery of ALK as a druggable oncogenic pathway in NSCLC led to the identification of an additional oncogene-addicted subgroup of NSCLC, for which the administration of ALK inhibitors completely changed the natural course of the disease.

Crizotinib is the pivotal ALK inhibitor that also inhibits ROS1 and MET receptor tyrosine kinases [78]. In the PROFILE 1001 [78] and PROFILE 1005 [79] phase I-II trials, crizotinib demonstrated good tolerability and showed an ORR of 59.8% (95% CI 53.6–65.9), a median PFS of 8.1 months (95% CI 6.8–9.7), and a median duration of response of 10.5 months. The PROFILE 1007 trial is the first phase III trial comparing crizotinib to standard second-line chemotherapy (docetaxel or pemetrexed), in patients with advanced *ALK*-rearranged NSCLC, progressing after a first-line platinum-based chemotherapy. The median PFS was significantly longer for crizotinib compared to chemotherapy (7.7 versus 3 months, HR 0.49; 95% CI 0.37 to 0.64; *p* < 0.0001) and ORR was strikingly in favor of crizotinib (65% vs. 20%; *p* < 0.001). The PROFILE 1014 trial compared crizotinib versus platinum-pemetrexed doublet in first-line setting and crizotinib still provided a significant improvement in PFS (10.9 vs. 7.4 months, HR 0.49, 95% CI 0.36 to 0.67; *p* < 0.0001). Both in the PROFILE 1007 and PROFILE 1014 [80,81], the OS analysis showed no significant improvement with crizotinib as compared with chemotherapy (in the PROFILE 1007 HR = 0.82, 95% CI 0.54–1.26, *p* = 0.36; in the PROFILE 1014 HR = 1.02, 95% CI 0.68–1.54, *p* = 0.54), but this result was inevitably affected by the crossover rate (70% of patients in chemotherapy arm). However, the adjusted HR for death ranged between 0.60 and 0.67 in favor of crizotinib. In consideration of the delay disease progression, upfront ALK TKIs therapy was approved in first-line [82]. 

Nevertheless, although the initial significant benefit, the efficacy of crizotinib, similarly to what happens with EGFR TKI in EGFR-mutant NSCLC, decreases after the onset of acquired resistance mechanisms, mainly secondary mutations in the ALK gene and activation of bypass tracks. Several innovative ALK inhibitors have recently demonstrated outstanding activity in both crizotinib-naïve and resistant ALK-rearranged NSCLC, with superior penetration of the BBB and RR on brain metastases.

Ceritinib is a second-generation, ATP competitive, highly selective ALK inhibitor, 20 times more potent than crizotinib against ALK protein kinases. Ceritinib is active against many ALK mutations that confer resistance to crizotinib, including C1156Y, F1174C, G1202R, and the gatekeeper mutation L1196M associated with high-level resistance to crizotinib. The ASCEND-4 and ASCEND-5 trials [82,83] demonstrated a statistically significant and clinically meaningful improvement in terms of PFS in favor of ceritinib versus chemotherapy respectively in TKI-naïve and in crizotinib pre-treated patients with advanced ALK-rearranged NSCLC. Nevertheless, in the alectinib era, few data are available regarding ceritinib post-alectinib with limited activity [84].

The current treatment algorithm radically changed with the introduction of alectinib, a potent, highly selective, second-generation ALK inhibitor which also showed high activity against L1196M, a common gatekeeper mutation leading resistance to crizotinib. The ALUR trial compared alectinib versus standard chemotherapy in advanced ALK positive NSCLC patients who have progressed on crizotinib. This trial demonstrated that PFS was significantly longer with alectinib (9.6 months [95% CI 6.9–12.2] versus 1.4 months [95% CI 1.3–1.6] with chemotherapy; HR 0.15) and also CNS ORR was significantly higher with alectinib (54.2%) versus chemotherapy (0%; *p*  <  0.001) [85]. ALESIA, J-ALEX, and ALEX phase 3 trials [86,87,88] compared alectinib versus crizotinib as first-line treatment for *ALK*-rearranged disease in Asiatic (ALESIA and J-ALEX) and in worldwide (ALEX) population. In both trials, alectinib showed superior efficacy and lower toxicity compared to crizotinib. In the J-ALEX trial, at the final PFS analysis, median PFS was 34.1 months (22.1–not estimated) with alectinib and 10.2 months (8.2–12.0) with crizotinib (HR 0.37; 95% CI 0.26–0.52). In the ALEX trial, PFS was significantly improved with alectinib versus crizotinib [HR 0.47 (95% CI, 0.34–0.65; *p* < 0.001]. Median PFS was 34.8 months (95% CI 17.7–not able to be estimated) with alectinib and 10.9 months (95% CI 9.1–12.9) with crizotinib (HR 0.43, 95% CI 0.32–0.58). OS data are still immature but alectinib appears to be superior. The most frequent adverse events in the alectinib group were anemia, diarrhea, nausea, liver dysfunction, myalgia, and edema. Given this advantage in terms of PFS, alectinib has been approved in the first-line in patients with *ALK*-positive NSCLC and became the current standard-of-care.

Brigatinib is another potent second-generation ALK inhibitor which showed a dual inhibition of ALK (including L1196M mutation conferring resistance to crizotinib) and EGFR T790M. The ALTA-1L trial [89] evaluated head-to-head the efficacy of brigatinib, as compared with crizotinib, in first-line in patients with advanced ALK-positive NSCLC. This trial showed that the rate of PFS was higher with brigatinib than with crizotinib (estimated 12-month PFS, 67% [95% CI 56–75] vs. 43% [95% CI 32–53]; HR 0.49 [95% CI 0.33–0.74]; *p* < 0.001). The ORR was 71% (95% CI 62–78) with brigatinib and 60% (95% CI 51–68) with crizotinib; the rate of intracranial response among patients with measurable lesions was 78% (95% CI 52–94) and 29% (95% CI 11–52), respectively [89]. The most frequent adverse events related to brigatinib were diarrhea, increased creatine kinase level, nausea, cough, and hypertension [89]. Comparing ALTA-1L and ALEX results, brigatinib and alectinib appear to have similar PFS and intracranial efficacy. With regard to safety, alectinib is associated with fewer grade ≥ 3 adverse events (41% with alectinib versus 61% with brigatinib) [88,89].

Lorlatinib is a novel, selective and potent ATP-competitive ALK and ROS1 inhibitor, with very high activity against EML4-ALK and all the recognized mutations driving resistance to crizotinib, ceritinib, and alectinib. In a phase I study, activity was seen in patients with ALK-positive NSCLC, most of whom had CNS metastases and progression after ALK-directed therapy [90]. In a multicohort phase II trial lorlatinib showed substantial overall and intracranial activity both in treatment-naïve patients and in those who had progressed on crizotinib, second-generation ALK tyrosine kinase inhibitors, or after up to three previous ALK tyrosine kinase inhibitors. Shaw et al. have demonstrated the lorlatinib antitumor activity against ALK G1202R/del, a mutation that confer resistance to both first- and second-generation ALK inhibitors [91]. In this study, lorlatinib was highly effective against ALK G1202R/del, with an ORR of 57%, median duration of response of 7.0 months, and median PFS of 8.2 months [92]. Thus, lorlatinib could represent an effective treatment option for patients with ALK-positive NSCLC in first-line or subsequent therapy to overcome acquired resistance mutations [93].

Although these drugs have radically changed the prognosis of this setting of disease, clinical outcomes in these patients vary. The presence of various breakpoints in the context of EML4-gene causes the formation of different variants of translocated EML4-ALK fusion proteins. These variants may be associated with the development of ALK resistance mutations, particularly G1202R, and provide a molecular link between variant and clinical outcome. [91].

Gainor J et al. presented the first study to evaluate mechanisms of resistance across a spectrum of first- and second-generation ALK inhibitors. Consistent with earlier reports, they found that only a minority of ALK-positive patients (~20%) developed ALK resistance mutations on crizotinib. By contrast, ALK resistance mutations were present in over one-half of patients progressing on second-generation ALK inhibitors, likely reflecting the greater potency and selectivity of these agents compared to crizotinib. In parallel, they observed that the spectrum of ALK resistance mutations was different following progression on second-generation ALK inhibitors compared to crizotinib. Most notably, ALK G1202R, which was present in only 2% of crizotinib-resistant biopsies, emerged as the most common ALK resistance mutation after treatment with second-generation ALK inhibitors [94]. While ALK G1202R was a common shared resistance mutation in each second-generation ALK inhibitor cohort, it is noteworthy that the spectrum of other ALK resistance mutations appeared to differ across agents. For example, ALK F1174 mutations were observed in several post-ceritinib biopsy specimens (4/24; 16.7%) but were otherwise absent from post-alectinib and post-brigatinib biopsies. Therefore, it will be important to incorporate repeat tissue or liquid biopsies into clinical trials of next-generation ALK inhibitors both before treatment and at progression. On a practical level, this work will also allow clinicians to personalize ALK-targeted strategies based upon the presence or absence of specific ALK resistance mutations, which may ultimately translate into improved patient outcomes [94] (Table 5).

There are many new trials coming out for ALK-rearranged NSCLC. One of the most attractive clinical trials currently recruiting is “The NCI-NRG ALK master protocol.” In this study, patients who have progressed on an ALK inhibitor will have a tissue genomic biopsy. This biopsy will determine the next treatment, based on the exact mutations in the progressed tumor. If the results of this study are better than the current standard of care, genomic testing upon progression, and individualized medicine could become the standard of care for ALK lung cancer progression [95].

## 4. ROS1 Rearrangements

The ROS1 oncogene encodes an orphan receptor tyrosine kinase related to ALK, along with members of the insulin-receptor family. ROS1 proto-oncogene receptor tyrosine kinase is activated by chromosomal rearrangement in a variety of human cancers, including NSCLC (approximately 1% of patients), cholangiocarcinoma, gastric cancer, ovarian cancer, and glioblastoma multiforme. Rearrangement leads to fusion of a portion of ROS1 that includes the entire tyrosine kinase. The resulting ROS1 fusion kinases are constitutively activated and drive cellular transformation [96,97].

The kinase domains of ALK and ROS1 share 77% amino acid identity within the ATP-binding sites. Crizotinib binds with high affinity to both ALK and ROS1, which is consistent with this homology. In the PROFILE 1001 trial, crizotinib showed marked antitumor activity in patients with advanced ROS1-rearranged NSCLC (ORR 72%, with a disease control rate equal to 90% and a median PFS of 19.2 months). On the basis of these results, crizotinib is currently recommended in the first-line setting for patients with advanced ROS1-rearranged NSCLC [78]. In phase 1–2 trials, ceritinib and lorlatinib demonstrated efficacy in this setting of disease (ORR 62% for both TKIs) with a great intracranial response rate (63% and 54%, respectively) and acceptable safety profile [98,99,100]. *Entrectinib* achieved high response rates (ORR 77%; CI 95% 64–88) and durable responses (median duration of response 24.6 months; 95% CI 11.4–34.8) in phase 1–2 trials, including in patients with brain metastases [101]. In the TRIDENT-1 trial, a phase 1–2 study, repotrectinib showed to be safe and active in patients with advanced ROS-1 NSCLC achieving ORR of 82% in TKI-naïve patients and 39% in TKI-pretreated patients. Repotrectinib was the only ROS1 inhibitor to show a potential to overcome TKI resistance mutations, notably G2032R, which is the most common ROS1 resistance mutation after crizotinib treatment [102]. With regard to these studies, it could be assumed that repotrectinib could become the treatment of choice in ROS1-rearranged patients to overcome G2032R resistance mutation. A larger sample size is necessary to validate these results.

## 5. BRAF Mutations

BRAF is an RAF kinase which is downstream of RAS and signals via the MAPK pathway. Signaling through RAF/MEK/ERK is complex, dependent on RAF dimerization, and limited by feedback inhibition of RAS signal. BRAF mutations are present in approximately 2–4% of lung adenocarcinomas, and the most common mutation is V600E [103]. BRAF mutations and other oncogenic drivers (EFGR, RAS, and ALK rearrangements) seem to be mutually exclusive [104]. The clinical outcome in patients with B V600E mutations is associated with shorter OS and lower response rates to platinum-based chemotherapy than in patients with BRAF wild-type. Prospective drug phase II trials of single-agent BRAF inhibitors (vemurafenib or dabrafenib) in patients with metastatic NSCLC and BRAF V600E show response rates of 30–40% and median PFS of 5–7 months [105,106]. Addition of the MEK inhibitor trametinib to dabrafenib in patients with BRAF V600E has shown an ORR of 63.2% (95% CI 49.3–75.6) in pretreated patients and 64% (95% CI 46–79) in naïve patients [107,108]. Given these results, the EMA and FDA have approved dabrafenib in combination with trametinib for the treatment of patients with BRAF V600 mutation-positive advanced or metastatic NSCLC.

## 6. MET Alterations

MET is a high-affinity proto-oncogene receptor tyrosine kinase that, upon activation, drives oncogenic pathways involved in cell proliferation, survival, and metastasis. Two major MET variants may play a role as drivers: Met exon 14 skipping and MET amplification. MET alterations leading to exon 14 skipping occur in ~4% of lung carcinomas, resulting in MET activation and sensitivity to MET inhibitors. Crizotinib demonstrated high activity in patients with MET exon 14 alterations (ORR 39% and median duration of response of 9.1 months) [96,109]. Capmatinib is a highly potent and selective MET inhibitor. In the GEOMETRY mono-1 trial, capmatinib has demonstrated a clinically meaningful ORR and a manageable toxicity profile in patients with METexon 14 skipping NSCLC, particularly in treatment-naïve patients where the ORR is 72% [110]. In preclinical study, capmatinib has also shown preclinical activity in combination with gefitinib in EGFR-mutant, MET-amplified/overexpressing models of acquired EGFR-TKI resistance. A phase Ib/II trial [111] showed that this combination could be a promising treatment for patients with EGFR-mutated, MET-dysregulated NSCLC, particularly MET-amplified disease (ORR of 47% in patients with a MET gene copy number ≥ 6). Tepotinib is another MET inhibitor that has shown promising activity in MET exon 14 skipping NSCLC with a favorable safety profile [112]. Further studies are needed to confirm these results. To date, crizotinib should be the treatment of choice for MET exon 14 altered NSCLC.

## 7. RET Mutations

The RET gene encodes a receptor tyrosine kinase that signals through the RAS/MAPK, PI3K/AKT, and JAK/STAT signaling pathways upon binding of native ligands [113]. Oncogenic RET gene rearrangements were first discovered in papillary thyroid cancer, primarily among patients exposed to ionizing radiation treatment or environmental radiation. The first RET gene fusion to be detected in lung cancer involved the 3′-end of RET and the 5′-end of the kinesin family member 5B gene (KIF5B). KIF5B-RET and its variants are the best characterized RET fusions in NSCLC, but at least 12 other fusion partners have been identified to date [114]. Similar to ROS1 fusions, the prevalence of RET rearrangements in NSCLC is approximately 1% and RET rearrangement-positive patients tend to be young and never smokers. While no RET-selective inhibitors are currently FDA approved, numerous multikinase inhibitors (vandetanib, lenvatinib, sunitinib) have been studied in patients with RET-rearranged lung cancer, but outcomes were modest when compared to the efficacy of TKIs targeting other NSCLC oncogenic drivers (median PFS is 2.3 months, and median OS is 6.8 months) [115]. Selpercatinib (LOXO-292) is a new highly selective RET inhibitor that has shown preclinical activity against a variety of RET fusions, as well as the RET V804M gatekeeper resistance mutation [116]. In the LIBRETTO-001 trial, LOXO-292 has shown marked antitumor activity in RET fusion-positive NSCLC, both in pretreated and treatment naïve patients ORR was 68% and CNS ORR was 91% in chemotherapy pretreated patients; ORR was 85% in treatment naïve patients. The safety profile was also favorable with a low discontinuation rate due to toxicity [117]. In the ARROW trial, patients with advanced RET fusion NSCLC were treated with BLU-667 achieving a high response rate (ORR 60% (95% CI 42–76) in chemo-pretreated patients) and great intracranial activity [118]. Targeting RET is not currently routinely recommended and recruitment into open trials is strongly encouraged.

## 8. NRTK Rearrangements

NTRK proteins play a vital role in the growth, differentiation, and apoptosis of neurons in the peripheral and central nervous stems. Chromosomal rearrangements of NTRK1 and NTRK 2 are known drivers of oncogenesis in several malignancies, including NSCLC, where the incidence is 3–4% [119]. Larotrectinib (LOXO-101) is a highly selective pan-TRK inhibitor that provided the first evidence of clinical benefit from TRK inhibition. Its efficacy was recently reported in a cohort of 55 patients with a variety of NTRK fusion-positive cancers, including four NSCLC patients. The outcomes across all the malignancies were promising with an ORR of 75% (median PFS had not yet been reached) [120]. Among the four NSCLC patients, three continued to have ongoing responses to the drug (ranging from 5.7 to 12 months), whereas one patient had stable disease [121]. These data highlight larotrectinib’s potent antitumor activity in NTRK fusion-positive patients. These findings led the FDA to grant accelerated approval of the drug for patients with solid tumors harboring an NTRK fusion without a known resistance mutation, making this the first pan-cancer approval of a TKI. The second-generation TRK-selective inhibitor, LOXO-195, has been used in two patients with NTRK fusion-positive cancer (colon and infantile fibrosarcoma) that developed resistance to larotrectinib. Both patients derived clear clinical benefits from the therapy [122]. Entrectinib (RXDX-101) is a first-generation pan-TRK inhibitor that also has activity against ROS1 and ALK fusions. Recently, a pooled analysis of ALKA-372-001, STARTRK-1, and STARTRK-2 trials reported the efficacy of entrectinib in 54 patients with NTRK fusion-positive solid tumors, revealing an ORR of 57.4%, median PFS of 11.2 months, and median OS of 20.9 months [123]. Based on these results, entrectinib and larotrectinib represent a rationale treatment choice in patients carrying NTRK rearrangements.

## 9. HER2 Aberrations

HER2 is a tyrosine kinase receptor of the ErbB/HER family and its amplification and overexpression represent a driver mutation in breast, ovarian, and gastric cancers [124]. HER2 aberration is found in 2–4% (mainly amplification and in-frame exon 20 insertions) of lung adenocarcinoma and it is one of the main mechanisms of acquired resistance to TKIs [125]. Targeted therapies such as dacomitinib, trastuzumab, and lapatinib have shown limited activity in NSCLC with HER2 alterations [126]. Afatinib and poziotinib have demonstrated some activity in HER2-mutated NSCLC [127,128]. Recently, two trastuzumab-based trials showed some promise for either the adotrastuzumab–emtansine conjugate (TDM-1) or trastuzumab/paclitaxel combination for HER2-positive lung cancers [129,130]. In vitro experiments have shown that osimertinib and pyrotinib have activity against NSCLC with HER2 exon 20 mutations [131,132].

To date, given the absence of robust data, targeting HER2 dysregulation is not currently recommended and there are no truly efficacy targeted therapies available. Therefore, recruitment into clinical trials is strongly recommended for these oncogene addicted diseases.

## 10. Conclusions

Undoubtedly, the advent of target therapy has radically changed the prognosis and the therapeutic paradigm of NSCLC patients. Within the last decade, much progress has been made in the identification of “driver” mutations and, consequently, of drugs that could delay tumor progression and markedly improve OS. Nevertheless, the percentage of patients with NSCLC carrying a driver mutation remains a minority. Certainly, further research is needed to identify further molecular targets and to develop new drugs to improve the prognosis and quality of life of these patients.

## Figures and Tables

**Table 1 cancers-12-01196-t001:** Immunotherapy in oncogene-addicted disease.

Study	Treatment	Patients	ORR, %	Gr> 3 AE	Median PFS, Mos	Median OS, Mos
TATTON Phase Ib	Osimertinib + Durvalumab	EGFR+ TKI naïve	70	47% (15% pneumonitis)	Not applicable	Not applicable
NCT 02088112 Phase I	Gefitinib + Durvalumab	EGFR+ TKI naïve	79	50% Mainly liver	Not applicable	Not applicable
NCT 02013219 Phase Ib	Erlotinib + Atezolizumab	EGFR+	75	39% Mainly pyrexia and liver	Not applicable	Not applicable
GEFTREM Phase I	Gefitinib + Tramelimumab	EGFR+ PD on TKI	67 stable disease	54% Mainly diarrhea	Not applicable	Not applicable
NCT 01454102 Phase I	Elotinib + Nivolumab	EGFR+	19	24% Mainly liver and diarrhea	Not applicable	Not applicable
Checkmate 370 Phase 1/2	Crizotinib + Nivolumab	ALK+	38	62% Mainly hepatitis and pneumonitis		
Nivolumab EAP Phase I	Nivolumab	EGFR+/ALK+	9	7%	3 (2.7–3.3)	8.3 (2.2–14.4)
IMPOWER 150 Phase 3	Atezolizumab + Carboplatin-Paclitaxel + Bevacizumab vs. Bevacizumab + Carboplatin − Paclitaxel	EGFR+/ALK+ PD on TKI	71	57%	10.2 vs. 6.9	NR vs. 18.7

AE: adverse events; MoS: months.

**Table 2 cancers-12-01196-t002:** Subgroup analyses of epidermal growth factor receptor (EGFR+) and anaplastic lymphoma kinase (ALK+) patients in large randomized control trials with immune checkpoint inhibitors.

Study	Treatment	Design	HR OS [95% CI]
KEYNOTE 010 Phase II/III	Pembrolizumab Docetaxel	PDL1+ NSCLC	0.88 (0.45–1.70)
POPLAR Phase II	Atezolizumab Docetaxel	NSCLC	0.99 (0.29–3.40)
Checkmate 057 Phase III	Nivolumab Docetaxel	Non-squamous NSCLC	1.18 (0.69–2.00)
OAK Phase III	Atezolizumab Docetaxel	NSCLC	1.24 (0.71–2.18)

**Table 3 cancers-12-01196-t003:** Differences between oncogene-addicted and non-oncogene addicted disease.

Characteristics	Oncogene Addicted Disease	Non-Oncogene Addicted Disease
Number of Drivers	Single (Dominant) Driver	Multiple Drivers and Passengers
Mutational Load Tumor Mutational Burden (TMB)	Small LOW TMB	Large HIGH TMB
Efficacy of Targeted Therapy (TKIs)	Yes, proven	No, still unproven
Efficacy of Immunotherapy	No, still unproven	Yes, proven
Early Resistance Rate	Low (≤20% at first evaluation)	High (≥50% at first evaluation)
Late Acquired Resistance (same/other pathways)	Always, proven	Few Late Acquired Resistance, unproven (long-term survivors, cured patients?)
Traditional Intermediate End-points Surrogacy (in absence of cross-over)	Yes	No

**Table 4 cancers-12-01196-t004:** Phase 3 comparative studies in the targeted therapy of EGFR mutant non-small-cell lung cancer (NSCLC).

Study	N	Treatment	ORR, %	Median PFS, Mos	Median OS, Mos
IPASS	261	Gefitinib vs. carboplatin/paclitaxel	71 vs. 47	9.8 vs. 6.4 (*p* < 0.001)	21.6 vs. 21.9 (HR: 1.00)
FIRST-SIGNAL	42	Gefitinib vs. cisplatin/gemcitabine	84 vs. 37	8.4 vs. 6.7 (*p* < 0.084)	27.2 vs. 25.6 (HR: 1.04)
NEJ002	230	Gefitinib vs. carboplatin/paclitaxel	74 vs. 31	10.8 vs. 5.4 (*p* < 0.001)	30.5 vs. 23.6 (HR: 0.89)
WJTOG 3405	172	Gefitinib vs. cisplatin/docetaxel	62 vs. 32	9.6 vs. 6.6 (*p* < 0.001)	34.8 vs. 37.3 (HR: 1.25)
OPTIMAL	165	Erlotinib vs. carboplatin/gemcitabine	83 vs. 36	13.1 vs. 4.6 (*p* < 0.0001)	22.8 vs. 27.2 (HR: 1.19)
EURTAC	174	Erlotinib vs. platinum-based chemotherapy	58 vs. 15	9.7 vs. 5.2 (*p* < 0.0001)	22.9 vs. 19.5 (HR: 0.93)
LUX-Lung 3	345	Afatinib vs. cisplatin/pemetrexed	56 vs. 23	11.1 vs. 6.9 (*p* = 0.001)	28.2 vs. 28.2 (HR: 0.88)
LUX-Lung 6	364	Afatinib vs. cisplatin/gemcitabine	67 vs. 23	11.0 vs. 5.6 (*p* < 0.0001)	23.1 vs. 23.5 (HR: 0.93)
LUX-Lung 7	319	Afatinib vs. Gefitinib	70 vs. 56	11 vs. 10.9 (*p* = 0.0073)	27.9 vs. 24.5 (HR: 0.86)
FLAURA	556	Osimertinib vs. Gefitinib	80 vs. 76	18.9 vs. 10.2 (*p* < 0.001)	38.6 vs. 31.8 (HR: 0.80)
ARCHER	452	Dacomitinib vs. Gefitinib/Erlotinib	74 vs. 71	14.7 vs. 9.2 (*p* < 0.0001)	34.1 vs. 26.8 (HR: 0.76)
NEJ 026	226	Erlotinib + Bevacizumab vs. Erlotinib	72 vs. 60	16.9 vs. 13.3 (*p* = 0.0157)	Not mature
JO25567	152	Erlotinib + Bevacizumab vs. Erlotinib	69 vs. 64	16 vs. 9.7 (*p* = 0.0015)	47.4 vs. 47 (HR: 0.81)
Cheng JCO	191	Gefitinib + Pemetrexed vs. Gefitinib	80 vs. 74	15.8 vs. 10.9 (*p* = 0.029)	43.4 vs. 36.7 (HR: 0.77)
NEJ009	344	Gefitinib + Pemetrexed-Carboplatin vs. Gefitinib	84 vs. 67	20.9 vs. 11.2 (*p* < 0.001)	52.2 vs. 38.8 (HR: 0.69)
RELAY	449	Erlotinib + Ramucirumab vs. Erlotinib	76 vs. 75	19.4 vs. 12.4 (*p* < 0.0001)	Not mature
Norohna et al. ASCO 2019	350	Gefitinib + Carboplatin/Pemetrexed vs. Gefinitib	75 vs. 62	16 vs. 8 (*p* < 0.0001)	NR vs. 17 (HR: 0.45)
ARTEMIS	311	Erlotinib + Bevacizumab vs. Erlotinib	157 vs. 154	18 vs. 11.2 (*p* < 0.001)	Not mature

**Table 5 cancers-12-01196-t005:** Phase 3 comparative studies in the targeted therapy of ALK-rearranged NSCLC.

Study	Therapy	Treatment	ORR, %	Median PFS, Mos	Median OS, Mos
PROFILE 1007	Second-line	Crizotinib vs. pemetrexed/docetaxel	65 vs. 20	7.7 vs. 3 (*p* < 0.0001)	20.3 vs. 22.8 (HR = 1.02)
PROFILE 1014	First-line	Crizotinib vs. cisplatin and pemetrexed	74 vs. 45	10.9 vs. 7.4 (*p* < 0.0001)	NR vs. 47.5
PROFILE 1029	First-line	Crizotinib vs. Cisplatin/Carboplatin and Pemetrexed	85 vs. 46	11.1 vs. 6.8 (*p* < 0.001)	28.9 vs. 27.7 (HR = 0.89)
ASCEND-5	Third-line	Ceritinib vs. pemetrexed/docetaxel	49 vs. 7	6.7 vs. 1.6 (*p* < 0.0001)	18.1 vs. 20.1 (HR = 1)
ASCEND-4	First-line	Ceritinib vs. Cisplatin/Carboplatin + Pemetrexed	72 vs. 27	13.5 vs. 6.7	NR vs. 26.2 (HR = 0.73)
ALUR	Second-line	Alectinib vs. docetaxel/pemetrexed	37 vs. 3 (*p* < 0.001) CNS ORR: 54 vs. 0	9.6 vs. 1.4 (*p* < 0.001)	12.6 vs. NR
ALESIA	First-line	Alectinib vs. Crizotinib	91 vs. 77 (*p*< 0.0001) CNS ORR: 73 vs. 22	NR vs. 11.1	Not mature
ALEX	First-line	Alectinib vs. crizotinib	83 vs. 75 (*p* = 0.09) CNS ORR: 86 vs. 71	34.8 vs. 10.9	Not mature
J-ALEX	First-line	Alectinib vs. crizotinib	85 vs. 70	34.1 vs. 10.2	Not mature
ALTA-1L	First-line	Brigatinib vs. crizotinib	71 vs. 60 *p* = 0.0678 CNS ORR: 78 vs. 29	NR vs. 9.8	Not mature

CNS: central nervous system; NR: not reached.

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
