# Peer review of "Oncogene-Addicted Non-Small-Cell Lung Cancer: Treatment Opportunities and Future Perspectives"

_cancers, 2020, doi:10.3390/cancers12051196_

Round 1
Reviewer 1 Report
The review manuscript by Ferrara is pertinent synopsis of the targeted anticancer therapies. It covers the critical molecular aberrations involving various kinases and kinase inhibitors targeting them, their mechanisms of kinase inhibition as well as mechanisms of resistance. It also catalogs their performance at the clinic with respect to such clinical outcomes as disease-free survival, overall survival periods before progression, etc. It is comprehensive yet succinct.
It does however appear as if this wasn’t their final version as some software-generated errors (lines 78, 80, 241 and 373) were not addressed. There are grammatical errors such as missing full stops (line 155), wrong tense (line 159), spelling (line 161), ‘was’ instead of ‘were’ (line 193), amino acid (line 391), ‘seems’ instead of ‘seem’ (line 413). The word ‘target’ instead of ‘targeted’ was also used (line, 41, 99, 242, 374, 485, 493 and 496). These minor corrections need to be made and table captions carefully examined for clarity.
Author Response
Dear Editor,
please find enclosed the revised manuscript ‘Oncogene-Addicted Non-Small-Cell Lung Cancer: Treatment Opportunities and Future Perspectives’ to be re-considered for publication in Cancers.
We want to particular thank the editor to give us the opportunity to deeply change the paper in order to be re-considered for publication. The suggestions have improved the value of the work done and our knowledge in this field, by critically pointing out the significant drawbacks.
As indicated, all the revisions suggested by the reviewers were performed as a point-by-point rebuttal, as follows:
Reviewer #1:
|
Point |
Reviewer’s Comment: |
Response: |
|
1 |
The review manuscript by Ferrara is pertinent synopsis of the targeted anticancer therapies. It covers the critical molecular aberrations involving various kinases and kinase inhibitors targeting them, their mechanisms of kinase inhibition as well as mechanisms of resistance. It also catalogs their performance at the clinic with respect to such clinical outcomes as disease-free survival, overall survival periods before progression, etc. It is comprehensive yet succinct. It does however appear as if this wasn’t their final version as some software-generated errors (lines 78, 80, 241 and 373) were not addressed. There are grammatical errors such as missing full stops (line 155), wrong tense (line 159), spelling (line 161), ‘was’ instead of ‘were’ (line 193), amino acid (line 391), ‘seems’ instead of ‘seem’ (line 413). The word ‘target’ instead of ‘targeted’ was also used (line, 41, 99, 242, 374, 485, 493 and 496). These minor corrections need to be made and table captions carefully examined for clarity. |
All the errors, mispelled words and typos have been revised. |
Additional comments:
- The affiliation of Dr. Vincenzo Di Noia was updated.
All authors have read and approved the revised manuscript. We confirm that each author has participated sufficiently in any submission to take public responsibility for its content. On behalf of all co-authors, we hereby certify that this manuscript is not currently under consideration and will not be submitted elsewhere until a final editorial decision has been reached.
We think that the changes made to the original manuscript address the main points raised by referee and that the revised manuscript will now be acceptable for publication.
Sincerely yours,
Prof. Emilio Bria, on behalf of all authors.

Reviewer 2 Report
Modify this review according to PRISMA.
Research manuscript sections are not respected.
Provide Material and Methods section to explain the review procedure.
It would be interesting to add a scheme of the therapeutic algorithm to summarise literature data.
The aim of the review is unclear.
Lines 78, 80, 240, 254, 373: Lack of reference source
Table 1: Specify PFS and OS in all the studies if applicable, otherwise specify that is not applicable.
Lines 110-112 Please clarify the sentence
Author Response
Dear Editor,
please find enclosed the revised manuscript ‘Oncogene-Addicted Non-Small-Cell Lung Cancer: Treatment Opportunities and Future Perspectives’ to be re-considered for publication in Cancers.
We want to particular thank the editor to give us the opportunity to deeply change the paper in order to be re-considered for publication. The suggestions have improved the value of the work done and our knowledge in this field, by critically pointing out the significant drawbacks.
As indicated, all the revisions suggested by the reviewers were performed as a point-by-point rebuttal, as follows:
Reviewer #2:
|
|
Reviewer’s Comment: |
Response: |
|
1 |
· Modify this review according to PRISMA. · Research manuscript sections are not respected. · Provide Material and Methods section to explain the review procedure. · It would be interesting to add a scheme of the therapeutic algorithm to summarize literature data. |
Given the nature of this review is narrative, the principles of a systematic review do not apply. For this reason, PRISMA workflow and the review process schema were not reported. Research manuscript sections are adapted to the manuscript type (review), therefore for example no 'material and methods' and 'results' paragraphs are required. In addition, the review is not intended to provide updated treatment algorithm or derive recommendation. That is why no therapeutic algorithms were provided in this narrative review. |
|
2 |
The aim of the review is unclear. |
A clearer sentence was added at the end of the abstract in order to clarify such point, as follows: ‘In this narrative review, the state of art of scientific literature about targeted therapy options in oncogene-addicted disease is summarized and critically discussed. We also aim to analyze future perspectives to maximize benefits for this subgroup of patients.’ |
|
3 |
Lines 78, 80, 240, 254, 373: Lack of reference source. |
Done. |
|
4 |
Table 1: Specify PFS and OS in all the studies if applicable, otherwise specify that is not applicable. |
Done. |
|
5 |
Lines 110-112 Please clarify the sentence. |
Done. |
Additional comments:
- The affiliation of Dr. Vincenzo Di Noia was updated.
All authors have read and approved the revised manuscript. We confirm that each author has participated sufficiently in any submission to take public responsibility for its content. On behalf of all co-authors, we hereby certify that this manuscript is not currently under consideration and will not be submitted elsewhere until a final editorial decision has been reached.
We think that the changes made to the original manuscript address the main points raised by referee and that the revised manuscript will now be acceptable for publication.
Sincerely yours,
Prof. Emilio Bria, on behalf of all authors.

Reviewer 3 Report
The authors herein report a review article on oncogene-addicted non-small cell lung cancer. Things have changes dramatically in last 2-3 years for NSCLC with sensitizing mutation. This review is the most complete, well written, exhaustive and very information. The paper can be improved in some ways:
- Abstract, line 24: The authors have mentioned that “oncogenic –addiction” more often occurs in, “in non-smoker”- This is not true for BRAF- mutated NSCLC. Please include this.
- Please include references for each of the study mentioned in the table.
- Section 1, line 97-100. Authors have indicated “studies” and only one “16” is cited. Please correct this part.
- Section 2, line 104-106. The prevalence of EGFR mutation is higher in Asian population, please mention that.
- Section 2, line 167: please mention the full form of EMA.
- Section 2, line 222-224: please include the reference [56] here at the end of sentence.
- Section 2, line 227-229: please include the reference for OS benefit. N Engl J Med. 2020 Jan 2;382(1):41-50
- The authors should also mention the preferred or current first-line treatment option (s) at the end of section for each mutation which will be very helpful information for the reader.
Author Response
All requested changes have been accoplished.

Round 2
Reviewer 2 Report
This is a simple review about already well known litterature data without insight and speculative ideas. The text doesn't follow author giudelines for reviews and additional informations added in this new version are usefull.
Author Response
We have reported all the required and applicable fields as reported in ‘Cancers Author Instructions’ (see below), by adding author contributions and conflicts of interest at 677-681 lines), except PRISMA guidelines that are not applicable in this narrative review, as already specified in Round 1 of this revision).
From Cancers Author Instructions. Reviews: These provide concise and precise updates on the latest progress made in a given area of research. Systematic reviews should follow the PRISMA guidelines. Review articles should be comprehensive and submitted by authors who are in the field. The main text of review papers should be around 4000 words and less than 6000 words with at least 2 figures or tables.
Review manuscripts should comprise the front matter (Title, Author list, Affiliations, Abstract, Keywords), literature review sections and the back matter (Supplementary Materials, Acknowledgments, Author Contributions, Conflicts of Interest, References).
